# Improvement in Subjective Symptoms and Tolerability in Response to Nintedanib Treatment in Elderly Patients with Idiopathic Pulmonary Fibrosis

**DOI:** 10.3390/jcm9030755

**Published:** 2020-03-11

**Authors:** Takayuki Takeda, Yusuke Kunimatsu, Nozomi Tani, Izumi Hashimoto, Yuri Kurono, Kazuki Hirose

**Affiliations:** Department of Respiratory Medicine, Japanese Red Cross Kyoto Daini Hospital, 355-5, Haruobi-cho, Kamanza-dori, Marutamachi-agaru, Kamigyo-ku, Kyoto 602-8026, Japan; ky92020223@yahoo.co.jp (Y.K.); nozomi-t@koto.kpu-m.ac.jp (N.T.); izumi-h@koto.kpu-m.ac.jp (I.H.); yuri.krn.cncl@gmail.com (Y.K.); k-hirose09@outlook.jp (K.H.)

**Keywords:** elderly patients, forced vital capacity, idiopathic pulmonary fibrosis, nintedanib, patient-reported outcomes

## Abstract

The efficacy of nintedanib treatment in patients with idiopathic pulmonary fibrosis (IPF) was demonstrated in phase III trials. However, there is limited data on the significance of nintedanib in elderly patients aged ≥75 years. We have retrospectively evaluated 54 newly nintedanib-treated patients including 32 elderly individuals. Potential changes in modified medical research council (mMRC) grade and COPD (chronic obstructive pulmonary disease) assessment test (CAT) score, as well as in forced vital capacity (FVC) were obtained 6 months before, at the time of, and 6 and 12 months after initiation of nintedanib treatment. Significant differences were observed in CAT scores between 6 months before treatment and baseline (*p* < 0.001), and between baseline and 6 months (*p* < 0.001) and 12 months (*p* < 0.001) after treatment. If subjective improvement is defined as an improvement in mMRC grade or CAT score by 1 or 3 points, respectively, 25 patients (46.3%) have significantly improved after 6 months of treatment. Out of these, all have improved in CAT score. The tolerability of nintedanib was similar in elderly and younger patients. These findings suggest that CAT scores could be useful in the subjective assessment during nintedanib treatment, and that nintedanib is safe and efficient for the treatment of the elderly population.

## 1. Introduction 

Idiopathic pulmonary fibrosis (IPF) is a type of chronic and progressive fibrosing interstitial lung disease of unknown etiology. It is distinct from other types by its progressive decline in forced vital capacity (FVC) and the poor prognosis with a median survival time of 2–5 years after diagnosis [1,2,3,4]. Even though lung function usually deteriorates gradually in patients with IPF, the clinical course is heterogeneous and unpredictable. Whereas some patients remain stable for years, others progress with a rapid decline in FVC [5,6], and the clinical course is sometimes interrupted by unpredictable acute exacerbation (AE) in IPF (IPF-AE) with high mortality [7,8,9,10]. Baseline features associated with increased risk of mortality in IPF [1] include the level of dyspnea, predicted diffusing capacity of the lung for carbon monoxide (DL_CO_) <40%, oxygen desaturation ≤88% during 6-min-walk test (6MWT), extent of honeycombing on high-resolution computed tomography (HRCT), and pulmonary hypertension. Additionally, a multidimensional index and staging system for IPF has been developed, using gender (G), age (A), and 2 lung physiology variables (P; FVC and DL_CO_), whereby IPF patients with GAP stage ≥2 are considered to have a poor prognosis [11]. Nevertheless, it remains very difficult to accurately predict the development of individual IPF patients.

Investigations into the molecular basis of IPF and the development of antifibrotic agents have revolutionized the treatment of this disease. Two antifibrotic agents, pirfenidone and nintedanib, have been confirmed to possess disease-modifying effects with an annual decline in FVC as well as in mortality through pooled phase III trials: CAPACITY-004 [12] and ASCEND [13] for pirfenidone and INPULSIS-1 and -2 [14] for nintedanib. Furthermore, it has been demonstrated that antifibrotic agents are applicable not only to relatively advanced IPF patients but also to those with preserved lung function [15,16]. In addition to the deterioration in FVC, the occurrence of IPF-AE should be considered, as the most frequent cause of death in patients with IPF is IPF-AE [3], which sometimes occurs even in early-stage patients who are regarded as stable [9]. Remarkably, a statistically prolonged survival has been observed over the last decade according to the European IPF registry, as the use of antifibrotic agents has increased to dominate IPF treatment [17]. Therefore, taking into account potential safety issues, appropriate antifibrotic treatment should be provided for every IPF patient from the time of diagnosis, regardless of lung function or GAP stage.

The number of IPF patients is increasing worldwide and the average age of onset in years is 71 (standard deviation [SD]; 11) in the UK [18], 73.5 (SD; 7.9) in the US [19], and 70 (SD; 9.0) in Japan [3]. Given the growing number of elderly patients with IPF and their elevated susceptibility to adverse events to pirfenidone, such as anorexia, dyspepsia, and hepatic dysfunction, as compared to younger counterparts [20], there is an urgent need for the development of a novel treatment strategy for this population. Indeed, nintedanib is considered less toxic [21], raising the likelihood of enhanced tolerability in elderly patients, which would allow early discontinuation of the antifibrotic agent to be circumvented. However, it remains unclear whether elderly patients with IPF receive the same benefit from nintedanib observed in INPULSIS trials. Therefore, the efficacy as well as safety of nintedanib in elderly patients should be elucidated.

Although an improvement in overall survival is important, it is not the only goal, since the IPF-associated symptoms greatly reduce the quality of life (QOL) of affected patients. The discrepancy between objective outcomes such as FVC and patient-reported outcomes (PROs) has been investigated in patients with incurable and life-shortening diseases [22], indicating that PROs could be a key means to optimizing antifibrotic treatment. The St. George’s Respiratory Questionnaire (SGRQ) score and the modified medical research council (mMRC) grade are usually applied as quantitative assessments of subjective symptoms in both usual care settings and clinical trials. However, while the SGRQ score is too complicated in the usual care setting, the mMRC is too simple for the assessment of the diverse symptoms of IPF. In contrast, the COPD (chronic obstructive pulmonary disease) assessment test (CAT) score has been proven suitable for the assessment of symptoms in patients with IPF [23]. 

Therefore, in the present study we used for the first time the mMRC and CAT as PROs, to evaluate the effects of nintedanib on subjective symptoms in elderly IPF patients. In addition to these tests, we examined the changes in FVC, focusing on potential differences in these 3 parameters between elderly and younger patients treated with nintedanib. The occurrence of potential adverse events of nintedanib were also explored in both groups.

## 2. Materials and Methods

### 2.1. Patients

A retrospective study was performed on 54 consecutively enrolled IPF patients newly treated with nintedanib at a dosage of 150 mg or 100 mg twice daily between November 2015 and October 2018. The diagnosis of IPF was based on the ATS/ERS/JRS/ALAT clinical practice guideline from 2015 [4]. Patients with possible usual interstitial pneumonia (UIP) pattern without honeycombing based on chest HRCT, and who met the INPULSIS trial eligibility criteria [14] (presence of reticular abnormality and traction bronchiectasis consistent with fibrosis with basal and peripheral predominance, and absence of atypical features for UIP), were clinically diagnosed as IPF/UIP. In such cases, findings that suggested temporal and/or spatial heterogeneity [24] were also taken into account. HRCT findings were confirmed by more than 2 radiologists. Patients who had been previously treated with nintedanib were excluded. Concomitant therapy with prednisolone up to 10 mg per day was permitted; however, patients receiving pirfenidone (concomitant as well as within 8 weeks before nintedanib introduction), *N*-acetylcysteine and/or other immunosuppressive agents including azathioprine, cyclosporine, or cyclophosphamide were excluded. This study was carried out in accordance with the Declaration of Helsinki and was approved by the Ethics Committee of the Japanese Red Cross Kyoto Daini Hospital (approval date: 21 June 2019; approval number: S2019-19). Written informed consent was omitted and the opportunity to opt out was provided on the homepage since the study was retrospective, and patient anonymity was assured.

The indication criteria for nintedanib at our hospital are as follows: FVC deterioration of more than 10% over 6 months; FVC deterioration of more than 5% over 6 months accompanied by symptoms; FVC deterioration of less than 5% but the patient requests nintedanib to prevent IPF-AE; and the presence of pulmonary fibrosis that is difficult to suppress by corticosteroids and/or immunosuppressive agents. 

The patients were classified into 2 groups according to age: (1) the elderly patients aged 75 years or older and (2) the younger patients under 75 years of age. The patients were numbered sequentially from 1 to 54, and individual data sets were managed according to the allocated number.

### 2.2. Subjective Assessments

Subjective symptoms and activities of daily living were monitored at every visit and were assessed quantitatively using the mMRC grade and CAT score. 

The CAT consists of 8 items: “cough”, “phlegm”, “chest tightness”, “breathlessness”, “activities”, “confidence”, “sleep”, and “energy” [25]. Each item is allocated a score from 0 to 5 so that the total score of 8 items ranges from 0 to 40, which shows good correlation with the SGRQ. The changes in mean mMRC grade and CAT score were evaluated every 6 months starting 6 months prior to nintedanib treatment initiation (also referred to as baseline), until 12 months thereafter. We evaluated the data of all participants and compared groups (elderly versus younger patients). The treatment was evaluated as “efficient” by subjective symptoms when the mMRC grade or CAT score improved by 1 or 3 points, respectively. 

### 2.3. Physiological Assessments

Lung function tests were performed in order to obtain objective FVC (mL and % predicted) every 6 months starting 6 months prior to baseline. DL_CO_ (% predicted) was obtained at the time of diagnosis to calculate GAP scores [11]. Oxygen desaturation during the 6MWT was also evaluated at the time of diagnosis.

The semiannual rate of change in FVC (%/6 months) prior to baseline was calculated by the difference between the FVC at baseline and that 6 months prior multiplied by 100 and divided by the FVC at baseline, which was defined as ∆FVC − 6M%. The semiannual (%/6 months) and annual (%/12 months) rates of change in FVC after beginning of nintedanib treatment were calculated by the difference between the FVC 6 or 12 months after baseline and the FVC at baseline (defined as ∆FVC + 6M and ∆FVC + 12M, respectively) multiplied by 100 and divided by the FVC at baseline, which were defined as ∆FVC + 6M% and ∆FVC + 12M%, respectively. When ∆FVC + 6M or ∆FVC + 12M were 0 or greater, they were considered stable or improved under nintedanib treatment, respectively.

### 2.4. Gender, Age and 2 Lung Physiology Variables (GAP) Stage

Points were assigned to each variable (gender, age, and physiology [FVC and DL_CO_]) to calculate a total score ranging from 0 to 8, and patients were classified as GAP stage I (0 to 3 points), stage II (4 to 5 points), and stage III (6 to 8 points) [11]. GAP stage was evaluated in each patient at the time of diagnosis. GAP total points (raw data) were utilized instead of GAP stages when comparing elderly with younger patients as well as in the analysis of factors associated to the changes in FVC and PROs.

### 2.5. Japanese Severity Stage

At the time of diagnosis, disease severity was evaluated in each patient according to the Japanese severity stage classification [26]. There are 4 stages according to the resting levels of arterial oxygen pressure (PaO_2_) and desaturation (SpO_2_ <90%) during 6MWT: stage I, PaO_2_ ≥80 mmHg; stage II, 80 > PaO_2_ ≥ 70 mmHg; stage III, 70 > PaO_2_ ≥ 60 mmHg; and stage IV, PaO_2_ <60 mmHg. In case of desaturation during 6MWT, the severity stage at II and III was increased by 1 stage.

### 2.6. Adverse Events

Adverse events were recorded according to the Medical Dictionary for Regulatory Activities version 20.1. Adverse events related to nintedanib between elderly and younger patients were evaluated.

### 2.7. Statistical Analysis

Data are presented as mean (standard deviation [SD]) or median (range) for continuous variables, and percentages (95% confidence interval [CI] or range) for categorical variables. The differences in background between 2 groups were evaluated using Fisher’s exact test for categorical variables, and Mann–Whitney U test for continuous variables. Differences in efficiency and adverse events were evaluated using Fisher’s exact test for the efficacy in PROs and adverse events, Mann–Whitney U test for the rates of change in FVC, and repeated measures ANOVA for the changes in FVC. In the analysis of factors that could affect the changes in FVC and PROs, multiple linear regression analysis and logistic regression analysis were applied, respectively. 

## 3. Results

### 3.1. Baseline Patient Characteristics

Baseline characteristics of this study are summarized in Table 1. The mean age was 74.5 years (SD; 4.9), and out of the 54 patients 7 (13.0%) were female and 47 (87.0%) were male. The following results were obtained for the mMRC grade: grade 0, *n* = 3 (5.6%); grade 1, *n* = 12 (22.2%); grade 2, *n* = 31 (57.4%); grade 3, *n* = 6 (11.1%); grade 4, *n* = 2 (3.7%). The mean CAT score was 12.85 (SD; 5.04). The starting dosage was reduced to 200 mg/day in 19 (35.2%) patients.

The mean CAT score was 12.78 (SD; 4.91) in elderly and 12.95 (SD; 5.33) in younger patients (*p* = 0.852). The starting dosage was reduced in 14 out of 32 (43.8%) elderly and in 5 out of 22 (22.7%) younger patients (*p* = 0.151). Significantly higher baseline GAP scores were observed in elderly (median; 5.0, range; 3.0–7.0) compared to younger patients (median; 4.0, range; 3.0–6.0) (*p* = 0.026), indicating a poorer expected prognosis in the former. There were no significant differences between groups in mMRC, Japanese severity stage, FVC, %FVC, existence of honeycombing or emphysema, previous pirfenidone treatment, or the reduction in the initial nintedanib dosage.

### 3.2. Therapeutic Effects of Nintedanib on Subjective Symptoms

Improvements in subjective symptoms (summarized in Table 2) were observed in 25 patients (46.3%) 6 months after nintedanib treatment, whereas 29 patients (53.7%) remained unchanged. Out of those who have improved, all had increased CAT scores by 3 points or more, while only 5 patients (20.0%) showed an increase in mMRC by 1 point. The subjective symptoms improved in 16 (50.0%) elderly patients and in 9 (40.9%) younger patients.

At 12 months after baseline, subjective improvements were the same overall (46.3%) and within groups (50.0% and 40.9% in elderly and young, respectively) as after 6 months. Subjective improvement was not associated with age or the existence of honeycombing or emphysema in either group.

Thus, CAT played an important and superior role in detecting subjective improvements in response to nintedanib treatment irrespective of age.

### 3.3. Therapeutic Effects of Nintedanib on Forced Vital Capacity

The mean FVC of all 54 patients 6 months prior to, at the time of, and 6 and 12 months after nintedanib treatment initiation were 2501.5 mL (SD; 600.3, range; 860–4090), 2286.3 mL (SD; 573.0, range; 770–3860), 2266.3 mL (SD; 548.4, range; 830–3700), and 2230.4 mL (SD; 550.1, range; 720–3710), respectively. As shown in Figure 1, in the elderly group, significant differences (worsening) were found between 6 months prior to baseline and baseline (*p* < 0.001), and between 6 and 12 months after baseline (*p* = 0.00684). Significant differences (worsening) within the younger group were observed between 6 months prior to baseline and baseline (*p* < 0.001), between baseline and 12 months after (*p* = 0.0161), and between 6 and 12 months after baseline (*p* = 0.0117). No significant differences in FVC changes were found between groups during the 12 months of nintedanib treatment.

The mean ΔFVC − 6M%, ΔFVC + 6M% and ΔFVC + 12M% in all 54 patients were −8.64% (range; −26.7–+2.11), −1.42% (range; −11.4–+12.2) and −2.66% (range; −11.7–+9.76), respectively (Table 2). The mean ΔFVC + 6M% was −1.18% (range; −11.1–+12.1) in elderly and −1.64% (range; −11.4–+12.2) in younger patients. The mean ΔFVC + 12M% was −1.36% (range; −11.6–+9.55) in elderly and –2.89% (range; −11.7–+9.76) in younger patients (Table 2). There was no significant difference regarding age or the existence of honeycombing and emphysema.

These results demonstrate the efficacy of nintedanib in reducing the decline in FVC during the 12-month follow-up and are in accordance with the phase III INPULSIS trials [14] and their post hoc analysis of the distribution of FVC changes [27]. 

### 3.4. Changes in Modified Medical Research Council (mMRC) Grades and Chronic Obstructive Pulmonary Disease Assessment Test (CAT) Scores

Among the elderly patients, the mean mMRC grades at each time point were 1.72 (SD; 0.73), 1.84 (SD; 0.85), 1.72 (SD; 0.73), and 1.72 (SD; 0.73), respectively (Figure 2A). The mean CAT scores in this group were 11.38 (SD; 4.99), 12.78 (SD; 4.91), 10.72 (SD; 4.43), and 10.66 (SD; 4.34), respectively (Figure 2B). 

In the younger patients, the mean mMRC grades at each time point were 1.82 (SD; 0.73), 1.86 (SD; 0.83), 1.82 (SD; 0.85), and 1.82 (SD; 0.85), respectively (Figure 2A). In this group, the mean CAT scores at each point were 11.68 (SD; 5.28), 12.95 (SD; 5.33), 11.09 (SD; 5.33), and 11.23 (SD; 5.57), respectively (Figure 2B).

While there was no significant difference in the mMRC grade among younger patients between each time point, significant differences were observed among elderly patients between 6 months prior to baseline and baseline (*p* = 0.0435), between baseline and 6 months after (*p* = 0.0435), and between baseline and 12 months after (*p* = 0.0435) (Figure 2A). On the other hand, CAT scores were significantly different in both elderly and younger patients between 6 months prior to and at baseline (*p* < 0.001), between baseline and after 6 months (*p* < 0.001), and between baseline and after 12 months (*p* < 0.001) (Figure 2B). There were no significant differences in subjective improvement between groups after 6 or 12 months of nintedanib treatment (*p* = 0.585 for both). Taken together, CAT scores provided a more detailed assessment compared with mMRC during nintedanib treatment, irrespective of age.

### 3.5. Relationship between Subjective and Objective Improvement

To understand better, the reasons behind the observed improvement in PROs, the relationship between PROs and FVC was examined further (summarized in Table 3).

In 17 and 13 patients, FVC values were stable or improved after 6 or 12 months of nintedanib treatment, respectively. In contrast, FVC values in 37 and 41 patients declined after 6 or 12 months, respectively. There was no significant correlation between ΔFVC + 6M or ΔFVC + 12M and mMRC grade. In contrast, 14 (82.4%) out of 17 patients with stable or improved ΔFVC + 6M also improved in CAT score, while only 11 (29.7%) out of 37 patients with declined ΔFVC + 6M showed such improvement (*p* < 0.001). Similar results were obtained for the 12-month follow-up, in which 10 (76.9%) out of 13 patients with stable or improved ΔFVC + 12M reported improvement in CAT score, whereas 15 (36.6%) out of 41 patients with declined ΔFVC + 12M showed such improvement (*p* = 0.023).

The changes in CAT score were in accordance with the clinical course of disease progression represented by changes in FVC (∆FVC + 6M and ∆FVC + 12M) by nintedanib treatment. These findings suggest that the CAT score could be useful in the subjective assessment of IPF during nintedanib treatment.

### 3.6. Idiopathic Pulmonary Fibrosis with Acute Exacerbation (IPF-AE)

IPF-AEs were observed in 2 (6.3%) elderly patients and 1 (4.5%) younger patient, but could be resolved by pulse corticosteroid therapy. There was no significant difference in the incidence of IPF-AE between groups (*p* = 1.000).

### 3.7. Adverse Events

In total, 40 patients (74.1%) experienced ≥1 adverse event. Adverse events included diarrhea in 31 patients (57.4%), nausea in 11 (20.4%), and elevation of aminotransferases in 14 (25.9%). No patient’s death as well as hospitalization was observed during the observation period. While 4 (12.5%) and 3 (9.4%) elderly patients and 2 (9.1%) and 1 (4.5%) younger patients had comorbidities of ischemic heart disease and cerebral infarction, respectively, which were stable and treated more than 12 months prior to the nintedanib introduction, no acute myocardial infarction or cerebral infarction were observed during the observation period. Diarrhea occurred in 17 (53.1%) elderly patients and 14 (63.6%) younger patients, nausea in 6 (18.8%) and 5 (22.7%), and elevation of aminotransferases in 8 (25.0%) and 6 (27.3%), respectively. There were no significant differences in adverse events or in the incidence of such between groups (*p* > 0.05), suggesting that the tolerability of nintedanib in elderly patients was the same as in younger patients.

### 3.8. Logistic Regression Analysis and Multiple Linear Regression Analysis

Logistic regression analyses were conducted to identify factors that may have affected the changes in PROs (raw data), FVC, and adverse events. Multiple linear regression analyses were applied to detect factors affecting the improvement in PROs (mMRC grade or CAT score improvement by 1 or 3 points, respectively).

While logistic regression analyses demonstrated that the existence of honeycombing and the baseline mMRC grade affected the changes in mMRC grade after 6 months of treatment (*p* = 0.0482 and 0.0188, respectively), no factors could be found that affected the changes in mMRC grade after 12 months. Moreover, smoking habits and a baseline mMRC grade ≥2 affected the changes in CAT scores after 6 and 12 months of treatment (*p* = 0.0367 and 0.0365 for 6-month time point, and *p* = 0.0150 and 0.0117 for 12-month time point, respectively).

Baseline GAP points (raw data) and FVC affected the changes in FVC after 6 months (*p* = 0.00737 and 0.00269, respectively), while there was no factor that affected the changes in FVC after 12 months.

Logistic regression analyses revealed that the reduced starting dosage of nintedanib (100mg twice daily) affected the incidence of diarrhea (*p* = 0.0426), and that baseline Japanese severity stage affected the incidence of nausea and elevation of aminotransferases (*p* = 0.0418 and 0.00538, respectively).

Multiple linear regression analyses demonstrated that the baseline Japanese severity stage also affected the improvement in PROs (mMRC grade and CAT score) after 6 months of treatment (*p* = 0.0125).

## 4. Discussion

In the present study, we found that out of 54 patients, 17 (31.5%) patients exhibited stable or improved FVC after 6 months of nintedanib treatment and FVC was sustained in 13 patients (24.1%) for at least 12 months. 

The PANTHER-IPF trial demonstrated that immunosuppressive treatment deteriorates the prognosis of IPF [28]. Experimental studies have elucidated that aberrant wound healing following various stimuli on alveolar epithelial cells, rather than an inflammation of the lung interstitium, is critical in the proliferation and differentiation of myofibroblasts [29], which is why antifibrotic treatment has replaced immunosuppressive treatment.

The introduction of pirfenidone or nintedanib at an early stage of IPF is widely accepted based on the results of phase III trials [12,13,14]. However, whether elderly patients could receive the same benefit from antifibrotic agents as observed in phase III trials has not been elucidated, with concerns over efficacy as well as tolerability. Our findings derived from subjective evaluations suggest that nintedanib exhibited favorable effects in elderly IPF patients, which is in accordance with the INPULSIS-1, -2 trials [14] and the INPULSIS-ON trial [30].

Regardless of the aforementioned clinical trials, in some cases antifibrotic treatment is initiated after the disease has already progressed. One reason could be the fear of adverse events, with a lack of clear and visible efficacy. While an increase in adverse events to pirfenidone has indeed been reported in elderly patients [20], the efficacy and tolerability of nintedanib in this population has not been investigated. Importantly, the results of the present study revealed that the adverse events in elderly patients were similar to previous reports [14] and that nintedanib was well tolerated.

The most reproducible and accessible way to measure IPF disease progression is the evaluation of lung function through FVC, and the assessment of FVC is widely accepted as a primary endpoint in clinical trials. Because the baseline FVC does not allow to predict the development of IPF and the FVC of patients with preserved lung function equally deteriorates over time [31], the follow-up evaluation of this parameter is of great importance. Since fewer than 25% of patients with preserved lung volume at baseline retained a stable FVC over the following year after diagnosis of IPF, antifibrotic treatment should be started irrespective of age or disease stage [16].

The underlying mechanism responsible for the improvement in FVC remains unclear. Nintedanib is a multikinase inhibitor that blocks platelet-derived growth factors and downstream signals, such as phosphatidylinositol-4,5-bisphosphate 3-kinase [32], which are important for cell growth and survival. Nintedanib has been reported to reduce the secretion of extracellular matrix (ECM) proteins such as collagen and to upregulate matrix metalloproteinases, which degrade excessive ECM [33]. The effects of nintedanib may result from the inhibition of further pulmonary fibrosis as well as the reduction of excessive ECM, thereby improving FVC.

Symptoms of IPF that may not be reflected by FVC function such as cough, dyspnea, and fatigue affect QOL and life expectancy and should, therefore, be fully considered in the treatment of IPF [34]. This could be achieved by focusing on subjective symptoms by means of PROs including mMRC grade and CAT score [35]. In the current study, we have shown that the CAT score could better reflect the subjective symptoms in elderly patients with IPF than the mMRC. The CAT scores improved equally in elderly patients compared with the younger ones during nintedanib treatment, which partly correlated with FVC improvement. It has been reported that the SGRQ total score is sensitive in the detection of FVC% predicted changes [36,37,38,39], which was also observed in the INPULSIS-2 trial [27]. Therefore, the evaluation of subjective symptoms by mMRC grade and CAT score could be useful in usual care settings. The correlation between improved ΔFVC + 6M or ΔFVC + 12M and the improvement in CAT score, which was observed in the current study, is consistent with previous reports and suggests that this correlation was applicable to elderly patients with IPF.

In addition to subjective symptoms and FVC, also IPF-AE should be considered with care, as it is the main cause of death in IPF patients [3]. Nintedanib has been shown to reduce the risk of IPF-AE [14,40]. Here, we found that this exacerbation occurred in 6.3% of elderly patients and in 4.5% of younger patients, which is not significantly different. Thus, considering the severity of IPF-AE and its risk factors [40,41,42,43], nintedanib treatment should be started early to reduce the risk of this exacerbation in all IPF patients.

The current study has some limitations. Since it was a retrospective observational investigation with a small sample size, the bias may be higher as compared to studies with larger populations. Therefore, the treatment strategy for elderly IPF patients with nintedanib should be confirmed in prospective randomized trials with larger sample sizes.

## 5. Conclusions

Taken together, we demonstrate the efficacy and tolerability of nintedanib in the treatment of elderly IPF patients. We also found that nintedanib exhibited positive effects on PROs, and that CAT score, also reflecting FVC changes, is superior to mMRC grading in assessing these effects, regardless of age. Seventeen and 13 patients out of 54 showed stable or improved FVC after 6 or 12 months of nintedanib treatment, respectively. The changes in CAT score were in accordance with these objective outcomes. Therefore, CAT scores may be useful in monitoring nintedanib treatment efficacy regardless of age.

## Figures and Tables

**Figure 1 jcm-09-00755-f001:**
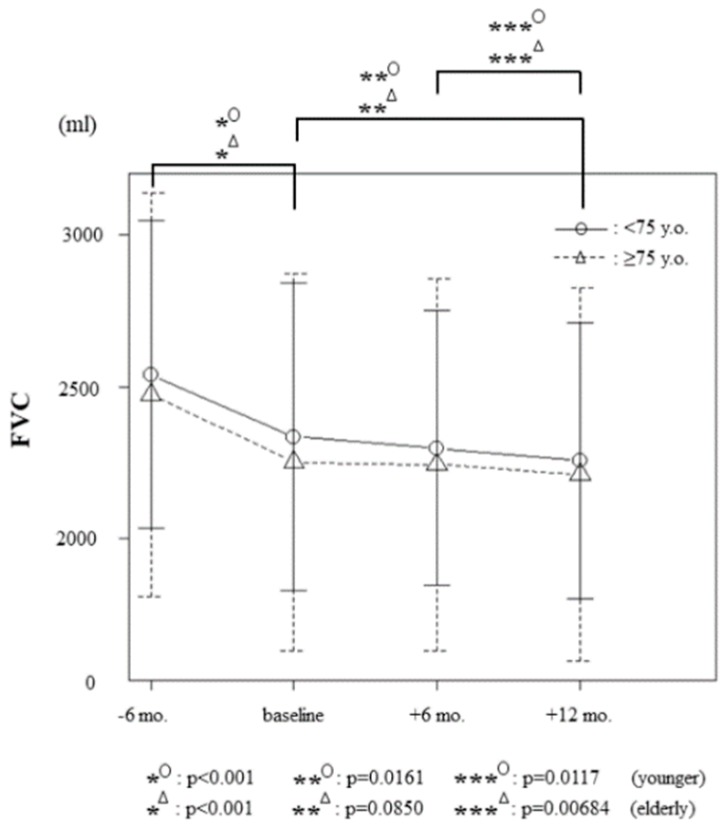
Changes in the forced vital capacity (FVC) 6 months prior to baseline, at baseline, and 6 and 12 months after initiation of nintedanib treatment. In the 32 elderly patients (≥75 y.o.), the mean FVC values at each time point were 2474.7 mL (standard deviation [SD]; 664.1, range; 860–4090), 2252.2 mL (SD; 620.9, range; 770–3710), 2244.4 mL (SD; 611.6, range; 830–3700), and 2211.9 mL (SD; 614.3, range; 720–3710), respectively. The 22 younger patients (˂75 y.o.) had mean FVCs of 2540.5 mL (SD; 505.7, range; 1980–4090), 2335.9 mL (SD; 505.2, range; 1750–3860), 2298.2 mL (SD; 452.7, range; 1670–3420), and 2257.3 mL (SD; 453.3, range; 1650–3410), respectively. The error bars show 2 SDs.

**Figure 2 jcm-09-00755-f002:**
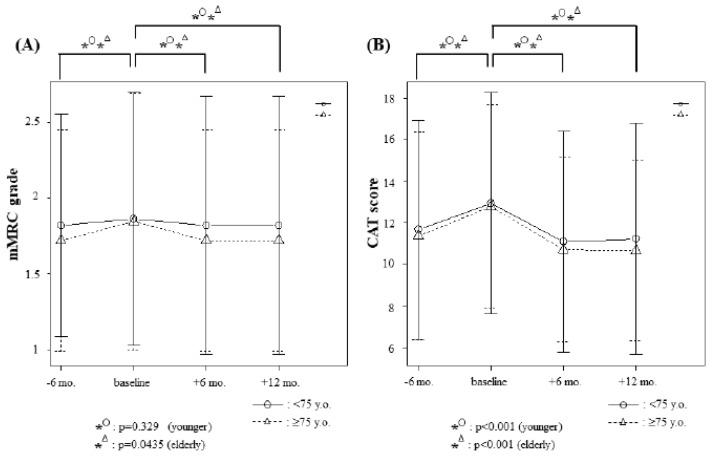
Changes in modified medical research council (mMRC) grade and COPD assessment test (CAT) score 6 months prior to baseline, at baseline, and 6 and 12 months after initiation of nintedanib treatment. In the 32 elderly patients (≥75 y.o.), the mean mMRC grades at each time point (**A**) were 1.72 (standard deviation [SD]; 0.73), 1.84 (SD; 0.85), 1.72 (SD; 0.73), and 1.72 (SD; 0.73), respectively. The mean CAT scores at each point (**B**) were 11.38 (SD; 4.99), 12.78 (SD; 4.91), 10.72 (SD; 4.43), and 10.66 (SD; 4.34), respectively. The 22 younger patients (˂75 y.o.) had mean mMRC grades (**A**) of 1.82 (SD; 0.73), 1.86 (SD; 0.83), 1.82 (SD; 0.85), and 1.82 (SD; 0.85), respectively. The mean CAT scores (**B**) were 11.68 (SD; 5.28), 12.95 (SD; 5.33), 11.09 (SD; 5.33), and 11.23 (SD; 5.57), respectively.

**Table 1 jcm-09-00755-t001:** Baseline Patient Characteristics of All 54 Patients.

	Total (*n* = 54)	≥75 years (*n* = 32)	<75 years (*n* = 22)	*p*-value
Mean age (years, SD)	74.50 (4.90)	77.75 (2.68)	69.86 (3.43)	<0.001
Sex (women / men; n, %)	7 (13.0%) / 47 (87.0%)	5 (15.6%) / 27 (84.4%)	2 (9.1%) / 20 (90.9%)	0.687
Never smoker (n, %)	20 (37.0%)	12 (37.5%)	8 (36.4%)	1.000
mMRC grade 0 / 1 / 2 / 3 / 4 (n, %)	3 (5.6%) / 12 (22.2%) / 31 (57.4%) / 6 (11.1%) / 2 (3.7%)	2 (6.2%) / 7 (21.9%) / 18 (56.3%) / 4 (12.5%) / 1 (3.1%)	1 (4.6%) / 5 (22.7%) / 13 (59.0%) / 2 (9.1%) / 1 (4.6%)	1.000
CAT score (mean, SD)	12.85 (5.04)	12.78 (4.91)	12.95 (5.33)	0.852
GAP stage I / II / III (n, %)	12 (22.2%) / 32 (59.3%) / 10 (18.5%)	3 (9.4%) / 22 (68.7%) / 7 (21.9%)	9 (40.9%) / 10 (45.5%) / 3 (13.6%)	0.026
Japanese severity stage I / II / III / IV (n, %)	9 (16.7%) / 25 (46.3%) / 18 (33.3%) / 2 (3.7%)	7 (21.9%) / 14 (43.8%) / 10 (31.2%) / 1 (3.1%)	2 (9.1%) / 11 (50.0%) / 8 (36.4%) / 1 (4.5%)	0.357
Mean FVC (ml, range)	2286.3 (770.0–3860.0)	2252.2 (770.0–3710.0)	2335.9 (1750.0–3860.0)	0.846
Mean % FVC (%, range)	72.2 (27.1–106.3)	72.8 (27.1–106.3)	71.8 (39.5–98.2)	0.470
Honeycombing + / − (n, %)	44 (81.5%) / 10 (18.5%)	26 (81.2%) / 6 (18.8%)	18 (81.8%) / 4 (18.2%)	1.000
Emphysema + / − (n, %)	26 (48.1%) / 28 (51.9%)	14 (43.8%) / 18 (56.2%)	12 (54.5%) / 10 (45.5%)	0.580
Previous pirfenidone (n, %)	12 (22.2%)	7 (21.9%)	5 (22.7%)	1.000
Nintedanib starting dose 300 / 200mg (n, %)	35 (64.8%) / 19 (35.2%)	18 (56.2%) / 14 (43.8%)	17 (77.3%) / 5 (22.7%)	0.151

Abbreviations: CAT, COPD assessment test; FVC, forced vital capacity; GAP, gender (G), age (A) and 2 lung physiology variables (P); mMRC, modified medical research council; SD, standard deviation. P-values refer to differences between groups by age.

**Table 2 jcm-09-00755-t002:** Therapeutic Effects of Nintedanib.

	Total (*n* = 54)	≥75 years (*n* = 32)	<75 years (*n* = 22)	*p*-value
Subjective Improvement after 6 Months (n, %)				
Elderly vs. Younger	25 / 54 (46.3%)	16 / 32 (50.0%)	9 / 22 (40.9%)	0.585
mMRC grade ≥ 1 point	5 / 54 (9.26%)	4 / 32 (12.5%)	1 / 22 (4.55%)	0.683
CAT score ≥ 3 points	25 / 54 (46.3%)	16 / 32 (50.0%)	9 / 22 (40.9%)	0.585
Honeycombing +/− ≥75y: *n* = 26 / 6, <75y: *n* = 18 / 4	(+); 21 / 44 (47.7%) (−); 4 / 10 (40.0%)	(+); 13 / 26 (50.0%) (−); 3 / 6 (50.0%)	(+); 8 / 18 (44.4%) (−); 1 / 4 (25.0%)	0.767 0.571
Emphysema +/− ≥75y: *n* = 14 / 18, <75y: *n* = 12 / 10	(+); 14 / 26 (53.8%) (−); 11 / 28 (39.3%)	(+); 7 / 14 (50.0%) (−); 9 / 18 (50.0%)	(+); 7 / 12 (58.3%) (−); 2 / 10 (20.0%)	0.713 0.226
Objective Improvement; Median ΔFVC%+6M (%, range)				
Elderly vs. Younger	−1.42 (−11.4 – +12.2)	−1.18 (-11.1 – +12.1)	−1.64 (−11.4 – +12.2)	0.398
Honeycombing +/− ≥75y: *n* = 26 / 6, <75y: *n* = 18 / 4	−1.36 (−11.4 – +10.8), −2.21 (−11.1 – +12.2)	−0.67 (−11.1 – +10.8) / −6.63 (−11.0 – +12.1)	−1.68 (−11.4 – +6.40) / −2.04 (−11.1 – +12.2)	1.000
Emphysema +/− ≥75y: *n* = 14/18, <75y: *n* = 12/10	−1.42 (−11.4 – +12.2), −1.59 (−11.1 – +10.8)	−1.10 (−11.4 – +12.1) / −1.17 (−11.1 – +10.8)	−1.10 (−11.4 – +12.2) / −2.74 (−11.7 – +9.76)	0.580
Subjective Improvement after 12 Months (n, %)				
Elderly vs. Younger	25 / 54 (46.3%)	16 / 32 (50.0%)	9 / 22 (40.9%)	0.585
mMRC grade ≥ 1 point	5 / 54 (9.26%)	4 / 32 (12.5%)	1 / 22 (4.55%)	0.683
CAT score ≥ 3 points	25 / 54 (46.3%)	16 / 32 (50.0%)	9 / 22 (40.9%)	0.585
Honeycombing +/− ≥75y: *n* = 26 / 6, <75y: *n* = 18 / 4	(+); 21 / 44 (47.7%) (−); 4 / 10 (40.0%)	(+); 13 / 26 (50.0%) (−); 3 / 6 (50.0%)	(+); 8 / 18 (44.4%) (−); 1 / 4 (25.0%)	0.767 0.571
Emphysema +/− ≥75y: *n* = 14 / 18, <75y: *n* = 12 / 10	(+); 14 / 26 (53.8%) (−); 11 / 28 (39.3%)	(+); 7 / 14 (50.0%) (−); 9 / 18 (50.0%)	(+); 7 / 12 (58.3%) (−); 2 / 10 (20.0%)	0.713 0.226
Objective Improvement; Median ΔFVC%+12M (%, range)				
Elderly vs. Younger	−2.66 (−11.7 – +9.76)	−1.36 (−11.6 – +9.55)	−2.89 (−11.7 – +9.76)	0.360
Honeycombing +/− ≥75y: *n* = 26 / 6, <75y: *n* = 18 / 4	−2.63 (−11.7 – +7.55), −2.66 (−6.61 – +9.76)	−1.94 (−11.6 – +7.84) / −4.08 (−6.61 – +9.55)	−2.51 (-11.7 – +5.91) / −1.63 (-5.75 – +9.76)	1.000
Emphysema +/− ≥75y: *n* = 14 / 18,<75y: *n* = 12 / 10	−2.67 (−11.7 – +9.76), −2.79 (−9.71 – +7.84)	−1.72 (−11.6 – +9.55) / −2.38 (−9.71 – +7.84)	−2.74 (-11.7 – +9.76) / −3.40 (-9.71 – −1.41)	0.151

Abbreviations: CAT, COPD assessment test; FVC, forced vital capacity; mMRC, modified medical research council; ∆FVC + 6M%, the semiannual rate of change in FVC after 6 months of nintedanib treatment; ∆FVC + 12M, the annual rate of change in FVC after 12 months of nintedanib treatment. P-values refer to differences between groups by age.

**Table 3 jcm-09-00755-t003:** Relationship Between Subjective and Objective Improvement.

**6-Month Evaluation**	**Stable or Improved** **ΔFVC+6M (*n* = 17)**	**Declined ΔFVC+6M** **(*n* = 37)**	***p*-value**
mMRC grade ≥ 1 point mMRC grade < 1 point	1 (5.9%) 16 (94.1%)	4 (10.8%) 33 (89.2%)	1.000
CAT score ≥ 3 points CAT score < 3 points	14 (82.4%) 3 (17.6%)	11 (29.7%) 26 (70.3%)	<0.001
**12-Month Evaluation **	**Stable or Improved** **ΔFVC+12M (*n* = 13)**	**Declined ΔFVC+12M** **(*n* = 41)**	
mMRC grade ≥ 1 point mMRC grade < 1 point	0 (0%) 13 (100%)	5 (12.2% )36 (87.8%)	0.321
CAT score ≥ 3 point sCAT score < 3 points	10 (76.9%) 3 (23.1%)	15 (36.6%) 26 (63.4%)	0.023

Abbreviations: CAT, COPD assessment test; FVC, forced vital capacity; mMRC, modified medical research council; ∆FVC+6M, the semiannual change in FVC after 6 months of nintedanib treatment; ∆FVC+12M, the annual change in FVC after 12 months of nintedanib treatment.

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
