# Peer review of "Improvement in Subjective Symptoms and Tolerability in Response to Nintedanib Treatment in Elderly Patients with Idiopathic Pulmonary Fibrosis"

_jcm, 2020, doi:10.3390/jcm9030755_

Round 1
Reviewer 1 Report
In the underlying study the authors show that the efficacy of nintedanib is similar in elderly patients as compared to younger patients using a retrospective analysis. Although the elderly patients are not necessarily excluded in the IMPULSIS and IMPULSIS-ON trial it is an interesting comparison showing once more that nintedanib is one of the best treatment strategies at this time for the treatment of IPF patients.
Furthermore the authors show as a secondary endpoint that CAT score is superior to mMRC grading.
The study is properly described and extensively analyzed. There are however also a few small limitations of this study.
As mentioned already by the authors, the number of patients included in the study is limited and if possible should be increased.
For a proper comparison with the original IMPULSIS trials similar inclusion and exclusion criteria should be used. In the underlying study patients that were previously treated with pirfinedone were included, whereas these were excluded in the IMPULSIS trials (within 8 weeks of screening). This group of patients makes over 20% of the cohort in the underlying study which is a significant limitation in the comparison. This should be recognized in the paper.
On page 2 the sentence starting with “In addition to the deterioration … regarded as stable.” Should be revised since it is grammatically incorrect.
Author Response
I really appreciate your sincere advice and suggestions and it is our great pleasure to be able to resubmit the revised manuscript.
We have read through your sincere suggestions and critiques, and prepared the revised manuscript that reflects your comments.
I would like to provide point-by-point responses below.
Comments and Suggestions for Authors
In the underlying study the authors show that the efficacy of nintedanib is similar in elderly patients as compared to younger patients using a retrospective analysis. Although the elderly patients are not necessarily excluded in the IMPULSIS and IMPULSIS-ON trial it is an interesting comparison showing once more that nintedanib is one of the best treatment strategies at this time for the treatment of IPF patients.
Furthermore the authors show as a secondary endpoint that CAT score is superior to mMRC grading.
The study is properly described and extensively analyzed. There are however also a few small limitations of this study.
- As mentioned already by the authors, the number of patients included in the study is limited and if possible should be increased.
Response:
We really appreciate your sincere advice. As you pointed out, the number of patients in the current study is limited due to a single institute retrospective study. The number of patients could be increased by redesigning the study to multicenter retrospective study.
However, this needs lots of procedure for the participants and their IRBs, which requires several months. Therefore, we would like to consolidate the current study in this sample size, and we would like to plan a multicenter retrospective study in the near future.
- For a proper comparison with the original IMPULSIS trials similar inclusion and exclusion criteria should be used. In the underlying study patients that were previously treated with pirfinedone were included, whereas these were excluded in the IMPULSIS trials (within 8 weeks of screening). This group of patients makes over 20% of the cohort in the underlying study which is a significant limitation in the comparison. This should be recognized in the paper.
Response:
We really appreciate your accurate assessment on our manuscript in detail.
As you pointed out, this is an important point.
We have checked that all the patients previously treated with pirfenidone were switched to nintedanib after more than 10 weeks of the cessation of pirfenidone.
Therefore, we inserted a description “…; however, patients receiving pirfenidone (concomitant as well as within 8 weeks before nintedanib introduction), N-acetylcysteine …” in the section 2.1. “Patients”.
- On page 2 the sentence starting with “In addition to the deterioration … regarded as stable.” Should be revised since it is grammatically incorrect.
Response:
We apologize for our incorrect grammar in the relevant sentence. We reworded the sentence according to your advice.
We changed the sentence to “In addition to the deterioration in FVC, the occurrence of IPF-AE should be considered, as the most frequent cause of death in patients with IPF is IPF-AE,3) which sometimes occurs even in early-stage patients who are regarded as stable.9)”
Reviewer 2 Report
The study answers an important question which is faced routinely in clinical practice- What is the efficacy of anti-fibrotic therapy in elderly population >75 years. The Study is conducted very methodically and explained well. Results are well presented, however can be expanded to help readers to understand the implications of anti-fibrotics better in elderly population.
- The study mentioned that Nintedanib was started in patients with FVC decline after 6 months of diagnosis. Can the authors explain why it was not started in all the patients at the time of diagnosis? Also, how many patients did not receive Nintedanib because their FVC did not decline.
- Please mention the co-morbidity data if available in these patients. There is a concern for increase in MI events in patients taking Ofev with coronary artery diseases.
- Please report deaths, hospitalizations during the study duration and IPF exacerbations.
- Were patients with emphysema diagnosed as IPF or combined pulmonary fibrosis and emphysema (CPFE?). This is important to explain because typically patients with CPFE tend to have worse prognosis as compared to patients with IPF.
Author Response
I really appreciate your sincere advice and suggestions and it is our great pleasure to be able to resubmit the revised manuscript.
We have read through your sincere suggestions and critiques, and prepared the revised manuscript that reflects your comments.
I would like to provide point-by-point responses below.
Comments and Suggestions for Authors
The study answers an important question which is faced routinely in clinical practice- What is the efficacy of anti-fibrotic therapy in elderly population >75 years. The Study is conducted very methodically and explained well. Results are well presented, however can be expanded to help readers to understand the implications of anti-fibrotics better in elderly population.
- The study mentioned that Nintedanib was started in patients with FVC decline after 6 months of diagnosis. Can the authors explain why it was not started in all the patients at the time of diagnosis? Also, how many patients did not receive Nintedanib because their FVC did not decline.
Response:
We really appreciate your suggestion.
As you indicate, it is important to start nintedanib or pirfenidone as soon as possible after the diagnosis of IPF. Of course, we advised almost all the patients who were diagnosed as IPF to treat with nintedanib or pirfenidone. However, most of the patients were diagnosed without symptoms with preserved lung function. This could be the reason why there were patients who did not receive nintedanib immediately after the diagnosis.
We introduced nintedanib according to “the indication criteria for nintedanib at our hospital” in the Materials and Methods.
As you pointed out, the number of IPF patients who did not receive nintedanib is of interest. However, the current study was performed on “54 IPF patients who were newly treated with nintedanib”. We need some more time to check all the patients who were diagnosed as IPF during the relevant period of time. And we also have to reconstruct the study design in order to expand the subject to “the patients who were newly diagnosed with IPF”, then we can further elucidate the number of IPF patients who were not treated with nintedanib because their FVC did not decline.
I asked the Ethics Committee of the Japanese Red Cross Kyoto Daini Hospital on this point, and I was told to undergo another IRB.
Then, I would like to preserve the current framework, and I will try to conduct other studies including IPF registry which would clarify your question.
- Please mention the co-morbidity data if available in these patients. There is a concern for increase in MI events in patients taking Ofev with coronary artery diseases.
Response:
We really appreciate your important suggestion and advice to improve our manuscript.
It is very important to show the comorbidity data and the adverse events in the analysis of elderly patients.
There were 4 (12.5%) elderly and 2 (9.1%) younger patients with stable ischemic heart disease treated more than 12 months prior to nintedanib introduction, and no patient resulted in acute myocardial infarction during the observation period. There were also 3 (9.4%) elderly and 1 (4.5%) younger patients with a past medical history of cerebral infarction treated more than 12 months prior, and no patient showed recurrence.
Since there no room for describing other comorbidities in Table 1, we added one sentence addressing this point in the part of Adverse Events.
We added a sentence “While 4 (12.5%) and 3 (9.4%) elderly patients and 2 (9.1%) and 1 (4.5%) younger patients had comorbidities of ischemic heart disease and cerebral infarction, respectively, which were stable and treated more than 12 months prior to the nintedanib introduction, no acute myocardial infarction or cerebral infarction were observed during the observation period” in the part of “3.7. Adverse Events”.
- Please report deaths, hospitalizations during the study duration and IPF exacerbations.
Response:
We are grateful for your advice. There was no patient who died or are hospitalized during the observation period.
As to those patients who developed IPF acute exacerbation, we reported 2 elderly patients and 1 younger patient in “3.6. IPF-AE” part.
We added a sentence “No patient’s death as well as hospitalization was observed during the observation period” in the part of “3.7. Adverse Events”.
- Were patients with emphysema diagnosed as IPF or combined pulmonary fibrosis and emphysema (CPFE?). This is important to explain because typically patients with CPFE tend to have worse prognosis as compared to patients with IPF.
Response:
We appreciate your suggestion and advice. As you pointed out, it is of scientific interest to distinguish those IPF patients with emphysema (combined pulmonary fibrosis and emphysema; CPFE) from emphysema patients who developed IPF thereafter.
However, it is difficult to distinguish them in the usual care setting.
I am sorry if I were wrong, my understanding is that the entity of CPFE is characterized by the presence of emphysema in upper zone and interstitial lung disease (diffuse parenchymal lung disease) in lower zone via computed tomography. Therefore, it could be very difficult for radiologists to distinguish the patients with CPFE from those with IPF who developed from COPD. And I understand that the prognosis of CPFE is worse than COPD, but better than IPF (Cottin V, Nunes H, et al. Eur Respir J. 2005; 26: 586-593.). This is one of the reasons why IPF patients with emphysema have been sometimes excluded from the clinical trials of antifibrotic agents, since the effect of the relevant antifibrotic agent could not demonstrate statistical efficiency when the prognosis of placebo group is not so bad.
And the coexistence of emphysema as well as its extent in patients with IPF reportedly did not affect the functional and prognostic outcome (Jacob J, Bartholmai BJ, et al. Eur Respir J. 2017; 50: 1700379.).
We included the data of emphysema in order to reconfirm the report in which the existence of emphysema did not influence the therapeutic effects of nintedanib in INPULSIS trials (Cottin V, Azuma A, et al. Eur Respir J. 2019; 53: 1801655.).
Therefore, I would like to let the relevant description of emphysema in the present form.